# Dissecting the Roles of Cuticular Wax in Plant Resistance to Shoot Dehydration and Low-Temperature Stress in *Arabidopsis*

**DOI:** 10.3390/ijms22041554

**Published:** 2021-02-04

**Authors:** Tawhidur Rahman, Mingxuan Shao, Shankar Pahari, Prakash Venglat, Raju Soolanayakanahally, Xiao Qiu, Abidur Rahman, Karen Tanino

**Affiliations:** 1College of Agriculture and Bioresources, University of Saskatchewan, Saskatoon, SK S7N 5A8, Canada; rahman.tawhidur@gmail.com (T.R.); mis706@mail.usask.ca (M.S.); prakash.venglat@gmail.com (P.V.); xiao.qiu@usask.ca (X.Q.); 2Saskatoon Research and Development Centre, Agriculture and Agri-Food Canada, Saskatoon, SK S7N 0X2, Canada; shankar.pahari@canada.ca (S.P.); raju.soolanayakanahally@canada.ca (R.S.); 3Department of Plant BioSciences, Faculty of Agriculture, Iwate University, Morioka, Iwate 0208550, Japan; abidur@iwate-u.ac.jp

**Keywords:** cuticular wax, dehydration, low temperature, freezing, stress avoidance, alkane

## Abstract

Cuticular waxes are a mixture of hydrophobic very-long-chain fatty acids and their derivatives accumulated in the plant cuticle. Most studies define the role of cuticular wax largely based on reducing nonstomatal water loss. The present study investigated the role of cuticular wax in reducing both low-temperature and dehydration stress in plants using *Arabidopsis thaliana* mutants and transgenic genotypes altered in the formation of cuticular wax. *cer3-6*, a known *Arabidopsis* wax-deficient mutant (with distinct reduction in aldehydes, n-alkanes, secondary n-alcohols, and ketones compared to wild type (WT)), was most sensitive to water loss, while *dewax*, a known wax overproducer (greater alkanes and ketones compared to WT), was more resistant to dehydration compared to WT. Furthermore, cold-acclimated *cer3-6* froze at warmer temperatures, while cold-acclimated *dewax* displayed freezing exotherms at colder temperatures compared to WT. Gas Chromatography-Mass Spectroscopy (GC-MS) analysis identified a characteristic decrease in the accumulation of certain waxes (e.g., alkanes, alcohols) in *Arabidopsis* cuticles under cold acclimation, which was additionally reduced in *cer3-6*. Conversely, the *dewax* mutant showed a greater ability to accumulate waxes under cold acclimation. Fourier Transform Infrared Spectroscopy (FTIR) also supported observations in cuticular wax deposition under cold acclimation. Our data indicate cuticular alkane waxes along with alcohols and fatty acids can facilitate avoidance of both ice formation and leaf water loss under dehydration stress and are promising genetic targets of interest.

## 1. Introduction

During their life cycle, plants are constantly exposed to a wide variety of changing environmental conditions and are able to adapt themselves through various mechanisms. To cope with the numerous stimuli generated by abiotic and biotic factors in the environment, plants have evolved complex structural, biochemical, and physiological processes, allowing them to withstand and complete their life cycle in their habitats. A key innovation in the plant evolutionary history was the development of the cuticle with its cuticular waxes, the first line of physical barrier against any external factors [1,2,3]. Several studies have shown cuticular waxes play an important role in the plant’s response to abiotic and biotic stresses [1,2,4,5]. Most importantly, its role in protection against nonstomatal water loss has been described extensively in both monocot and dicot plant species [2].

Cuticular waxes are composed of complex mixtures of hydrophobic very-long-chain fatty acids (VLCFAs) and their derivatives, including alkanes, esters, alcohols, alkenes, aldehydes, and ketones [2,3,6,7]. There is a vast diversity in the structure and composition of cuticular waxes in different plant species [2,3,8,9], within different plant organs, and at different stages of plant development [8,9]. In *Arabidopsis*, cuticular wax layers in the cuticle are mainly composed of alkanes (>70%) with lower presence of alcohols, aldehydes, fatty acids, and ketones [1]. The genetics of cuticular wax synthesis has been extensively studied in *Arabidopsis* and other crops. A plethora of genes have been identified that play roles in perceiving various stresses, regulate wax biosynthesis and its deposition in the cuticle in a quantitative and qualitative manner, and are spatially localized to different plant organs [1,2,3]. Despite the large number of wax-related genes which have been cloned and characterized, only a few have been assessed for their direct role against stresses. Members of ECERIFERUM *(CER)* genes involved in wax biosynthesis (e.g., *CER1*, *CER3*, *CER6*) and transcription factor (TF) genes (e.g., *WIN1* or *SHN1*, *DEWAX*, *MYB94*, *MYB96*) regulating the expression of *CER* genes have been characterized for their roles in drought-resistance mechanisms [1,2,3,9,10]. *Arabidopsis CER1* and *CER3* participates in the acyl decarbonylation pathway to produce alkanes. *CER3* is proposed to encode a VLCFA reductase, catalyzing the formation of fatty aldehydes, and CER1 functions as an acyl decarbonylase, catalyzing the formation of alkanes from fatty aldehydes [11,12,13]. In fact, biosynthesizing VLC alkanes through the acyl decarbonylation pathway is an integral part of cuticular wax biosynthesis in all monocot and dicot plant species.

Expression of *CER1* and *CER3* occurs in the epidermis of aerial organs in response to abscisic acid (ABA), drought, and osmotic stresses [2,10,14]. *CER1*-overexpressing plants possessed 24–32% greater total wax content (with a 87–101% increase in the amount of alkanes) and increased resistance to drought stress [10]. Conversely, *cer1* mutant lines showed reduced wax content (44–56% lower in wax load per unit leaf area) compared to the parental wild-type (WT) genotype [10]. The *cer3* (also known as *wax2*) mutant of *Arabidopsis* was found to be deficient in alkanes, secondary alcohols, aldehydes, and ketones by 78–83%, and exhibited elevated sensitivity to dehydration water loss as compared to the parental genotype [14,15,16]. CER3 is required for both cuticular wax and cutin deposition, and the protein was shown to exhibit 32% similarity to CER1 in *Arabidopsis* [14]. The cucumber homolog of the *CER3* gene, *CsWAX2,* is induced by drought, ABA, low temperature, and salinity. In comparison to the WT, *CsWAX2*-overexpressing cucumber plants showed significant improvement in resistance to drought and pathogens [17]. Heterologous expression of *Arabidopsis CER1* and *CER3* in transgenic tomato lines conferred increased epicuticular wax, predominantly consisting of alkanes, and had reduced water loss and higher water use-efficiency (WUE), with zero reduction in biomass under drought stress [2]. These results suggest *CER3* and *CER1* are evolutionarily conserved genes in plants playing an important role in the biosynthesis of wax and modulating the resistance to stresses.

*CER4,* another conserved wax biosynthetic gene encoding an alcohol-forming fatty acyl-coenzyme A reductase (FAR), catalyzes the synthesis of long-chain primary alcohols in the epidermal cells of aerial tissues and in roots [18]. *Arabidopsis cer4* mutant displays reduced levels of primary alcohols and wax esters, and slightly elevated levels of aldehydes, alkanes, secondary alcohols, and ketones [18]. The *CER6* gene, also known as *CUT1* or *KCS6*, encodes a member of the 3-ketoacyl-CoA synthase family involved in the biosynthesis of VLCFA longer than C28. The suppression of *CER6* expression resulted in a dramatic decrease of waxes longer than C24 in the stem [19]. It was found that mutant allele *cer6-1* displayed a phenotype similar to *cer1* and *cer3* (phenotypes curated at the *Arabidopsis* Biological Resource Center). *Arabidopsis CER2* was found to have sequence similarity to a BAHD acyltransferase involved in elongation of VLCFA waxes longer than C28 in the stem cuticle, while *CER2-LIKE1*, expressed predominantly in the leaves, was found to be functionally redundant to *CER2* [20]. A double mutant *cer2-5 cer2-like1* showed deficiency of C31 alkane and C30 primary alcohol in the rosette leaf cuticle with a total leaf wax load similar to parental line [20].

Previous studies have shown that expression of several TF genes with a role in cuticular wax biosynthesis were upregulated under various environmental stresses [1,21,22,23]. *Arabidopsis SHNs* (e.g., *SHN1, SHN2,* and *SHN3*) encode proteins of the APETALA 2/Ethylene Response Element Binding Protein (AP2/EREBP) family of TFs. Overexpression of *SHN1* resulted in altered epidermal properties and increased epicuticular wax and drought resistance in *Arabidopsis* [24,25]. Heterologous expression of *SHN1* in transgenic mulberry conferred higher resistance to drought and displayed similar phenotypes to *SHN1*-overexpressing *Arabidopsis* lines [26]. Additionally, the AP2/EREBP TF *DEWAX* has been shown to function as a negative regulator of wax synthesis in *Arabidopsis*, with the *dewax* mutant displaying greater accumulation of cuticular wax in the leaf and stem compared to WT [27]. Further, overexpression of *DEWAX* reduced the expression of various wax biosynthetic genes in *Arabidopsis*. In addition, two MYB family transcription factor genes, *MYB94* and *MYB96*, induced the wax biosynthetic pathway in *Arabidopsis* [28,29]. Both genes were significantly upregulated in response to ABA, drought, and salt, and a significant decrease in the cuticular wax load was observed in *myb94 myb96* double mutant [28]. An activation tag line of MYB96 and an overexpressing line of MYB94 showed dramatic increase in the cuticular wax load, further confirming the role of these two TFs in wax biosynthesis [23,28,29].

Imbalance in water loss is the most common physiological phenomena under almost all abiotic stresses. Drought treatments increased the total wax load and altered the composition of cuticular wax in *Arabidopsis* [1,23,24], *Triticum* [30,31], *Pisum* [32], *Medicago* [22], and *Rosa* [33]. Specifically, in leaf tissues, drought stress increased cuticular wax deposition by up to 2.5-fold in a number of plant species, including *Zea*, *Triticum*, *Glycine*, *Pinus*, *Avena*, *Nicotiana*, *Gossypium*, and *Sesamum* [2,4,34]. Of the different wax components, very-long-chain (VLC) n-alkanes (C29–33) constituted almost 93% of the total wax in *Arabidopsis* under prolonged water deprivation [35]. Reduction of alkane monomers, especially the C29 alkane, was shown to reduce plant tolerance to drought stress [21]. These studies suggest that alkane plays a key role among all wax components in the cuticle against drought stress.

An extensive physiological and genetic evidence exists not only regarding the drought-mediated changes of the cuticular wax in plants, but also the role of wax in imparting drought resistance. In comparison, there is a lack of information regarding the role of plant cuticular wax in frost avoidance. The leaf cuticular surface was shown as the first barrier blocking destructive ice penetration into the leaf cells during freezing events [36]. The importance of the cuticular hydrophobicity enabling avoidance of freezing in sensitive plants was shown using a hydrophobic film [37]. Since various wax constituents in the plant cuticle primarily contribute to leaf hydrophobicity, it is likely that changes in the wax composition could alter the ability to avoid freezing in plant species.

This study sought to understand the changes in cuticular waxes in response to low-temperature and examined how those changes contribute to the freezing avoidance in *Arabidopsis*. In the present study, we investigated a set of *Arabidopsis* wax-deficient, wax-overproducing mutants and transgenic genotypes previously characterized for gene-specific functional studies. Our results revealed that *cer3-6* mutant, deficient in wax production, showed reduced ability to avoid freezing following cold acclimation and increased leaf water loss under dehydration stress. In contrary, the wax-overproducing *Arabidopsis dewax* mutant displayed an opposite phenotype to *cer3-6* under the same freezing and dehydration treatments, indicating the importance of cuticular wax in proper acclimation and freezing resistance. Consistently, Gas chromatography Mass spectroscopy (GC-MS) and Attenuated Total Reflection Fourier Transform Infrared Spectroscopy (ATR-FTIR) studies confirmed significant alteration in the cuticular wax formation in *Arabidopsis* under cold acclimation. Cold acclimation decreased the conversion of VLCFA to more hydrophobic constituents such as VLC primary alcohols or VLC alkanes, while the unconverted VLCFAs were increased in the cuticle. Taken together, these results suggest that CER3 plays a critical role for biosynthesis of important hydrophobic wax constituents required to avoid frost and dehydration stress in *Arabidopsis*.

## 2. Results

### 2.1. Higher Accumulation of Cuticular Wax Does Not Inhibit the Reproductive Yield in Arabidopsis

All *Arabidopsis* genotypes used in the study were monitored visually for their growth and development under control growth conditions (22 °C/20 °C), 16/8 h light-dark cycle at 100 µE m^−2^ s^−1^) before starting the stress experiments. Humidity in the chamber was not controlled. Siliques were observed for quantity of normal/abnormal siliques and the presence/absence of seeds. Although there was no germination or early developmental issues with any of the genotypes, abnormal silique formation phenotypes were observed in certain *cer* mutants. The mutant alleles *cer1-4*, *cer3-6*, and *cer6* produced higher numbers of siliques than WT (Figure 1A,B). On average, *cer1-4*, *cer3-6*, and *cer6* produced 38%, 121%, and 116% more siliques at maturation as compared to WT. However, compared to WT, almost all the siliques in *cer1-4*, *cer3-6*, and *cer6* lacked seeds. In comparison to the *cer* mutants, *dewax*, *cer2-5 cer2-like1*, and MYB94 OX produced morphologically normal siliques with intact seeds similar to WT, with *dewax* mutant producing 42% more siliques with seeds compared to WT. The *DEWAX* overexpressing lines, DEWAX OX1 and OX2, displayed certain abnormal silique phenotype to the *cer* mutants (e.g., *cer1-4*, *cer3-6*, and *cer6*), producing many siliques without any seeds.

### 2.2. The Mutant Allele cer3-6 Is Highly Sensitive to Shoot Dehydration

To investigate the performance of wax-deficient and overproducing lines, we initiated a whole-shoot dehydration assay with plants grown under control conditions for three weeks. At 1-min intervals over a period of 20 min, excised shoots sealed at the cut end were monitored for weight loss (Figure 1C). Mutant allele *cer3-6* lost weight at a much faster rate (~0.75%/min) compared to WT (~0.50%/min). MYB94 OX, *cer2-5 cer2-like1*, and *dewax* displayed reduced dehydration water loss compared to WT (Figure 1C). The average weight loss per minute for MYB94 OX, *cer2-5 cer2-like1*, and *dewax* was 0.19%, 0.38%, and 0.4% per minute, respectively. *cer1-4* had a similar rate of water loss to WT. The *cer6* mutant lost weight at 0.6% per minute. Consistent with the water loss-resistant phenotype of the *dewax* mutant, the overexpressing line DEWAX OX1 showed sensitivity to water loss with an average of 0.63% weight loss per minute.

### 2.3. Cold Acclimated cer3-6 and Dewax Display Contrasting Ice Nucleation at Warmer Subzero Temperatures

To investigate the role of cuticular wax in plant responses to freezing stress, mutants and overexpressing lines were exposed to subzero temperature treatment following a 2-week of cold acclimation regime of 5 °C/2 °C under a 16/8 h light-dark cycle. Under an extracellular freezing tolerance test in which plants were first nucleated at −3 °C and temperatures slowly dropped to −6 °C, −10 °C, −14 °C, and −18 °C, after 7 days of recovery, there were no significant differences in survival among all the genotypes (data not shown).

The two mutant alleles, *cer3-6* and *dewax*, displayed consistently differential phenotype in response to rapid freezing treatment with no artificial ice nucleation. Cold-acclimated *cer3-6* plants initiated freezing in the rosette leaves at ~1.65 °C-warmer temperatures compared to the cold-acclimated WT seedlings. In comparison, cold-acclimated *dewax* seedlings resisted freezing for a longer period of time and initiated ice nucleation at ~1.5 °C-colder temperatures than WT (Figure 2A,B). Freezing experiments for cold-acclimated *cer3-6* and *dewax* were also conducted separately with the cold-acclimated WT seedlings. The resulting infrared thermography data for cold-acclimated *dewax* and *cer3-6* relative to cold-acclimated WT displayed the same trend (Table 1). The cold-acclimated *cer1-4* mutant generally froze at temperatures between WT and *cer3-6*, but since data were variable, it was not included in all repeated freezing experiments.

### 2.4. Accumulation of Hydrophobic Wax Deposition in the Cuticle Decreases under Cold Acclimation

To investigate the changes in the composition of cuticular wax under cold acclimation, GC-MS analysis was performed with wax samples extracted from non-acclimated and cold-acclimated WT, *cer3*, and *dewax*. Data and statistical analysis are presented in Figure 3A,B and in Table 2. In the non-acclimated controls, *cer3-6* had the lowest levels in all three categories of fatty acids, alkanes, and alcohols (Figure 3A,B), creating a significant interaction between genotype and treatment in C28 fatty acids, C31 and C35 alkanes, and C29 alcohols. Compared with WT, *cer3-6* exhibited the lowest quantity of C26 and C30 fatty acids, while *dewax* showed significantly greater fatty acids (C24, C32, C34) (Figure 3A,B). In terms of alkanes, *dewax* and WT generally were not significantly different. However, the most distinct difference was observed in the lowest alkane levels in *cer36* (C29, C31, C33). Alcohols were greater in *dewax* than WT in the C32 and C34 chain lengths, with *cer3-6* significantly lower than WT and *dewax* in this category (Figure 3A). After cold acclimation, levels of VLC fatty acids (C24–C34) were increased, whereas the levels of alkanes (C29–C35) and alcohol waxes were decreased in all three genotypes (Figure 3A,B), with alkanes falling by 94% in *cer3-6* after acclimation. Again, *cer3-6* had the lowest relative levels in all three categories of fatty acids, alkanes, and alcohols compared to WT and *dewax*. However, the most striking shift was in the acclimation-induced increase in alkane levels in *dewax* compared to WT, where there was no difference under non-acclimated conditions. The *dewax* line maintained a significantly higher level of alkanes in the rosette leaves during acclimation as comparted to WT (*dewax* accumulated 17 µg more alkanes/cm^2^ leaf than WT under the control growing environment, doubling to 35.58 µg/cm^2^ leaf under cold acclimation).

### 2.5. ATR-FTIR Analysis of Leaf Epidermal Surfaces Identify Changes in Lipid Accumulation in Response to Cold Acclimation

Nondestructive ATR-FTIR analysis of the rosette leaf cuticle was conducted primarily to understand the biochemical changes of wax lipids in response to cold acclimation in the *Arabidopsis* wax mutant and overproducing lines included in the study. FTIR spectral profile of the leaf epidermal surface generated characteristic spectra both in the main lipid region (3034–2819 cm^−1^) and in the fingerprint region (1800–800 cm^−1^) in all genotypes, although the data showed differential band intensity areas in mutants and WT under control and acclimation conditions (Figure 4A and Appendix A). The peak area intensities of CH_3_ stretching (CH_3_), CH_2_ symmetric (CH_2_s), and CH_2_ asymmetric (CH_2_a) of the main lipid region were increased in most of the genotypes in response to cold acclimation. In the case of the *cer3-6* mutant, no changes under cold acclimation were observed. However, the CO stretching, which had a characteristic peak at ~1736 cm^−1^, displayed a sharp increase in *cer3-6* mutant compared to WT and *dewax* (Figure 4A). Specifically, the CO stretching was increased by 20%, 50%, and 15% in WT, *cer3-6,* and *dewax*, respectively, in response to cold acclimation (Table 3). In general, the band intensities of CH_2_s and CH_2_a were higher in the wax-overproducing line *dewax* than WT under both control and cold acclimation conditions. CH_3_ peak areas were increased by 48%, 15%, and 45%, whereas CH_2_a peak area were up by 70%, 8.5%, and 73% in WT, *cer3-6*, and *dewax*, respectively, in response to cold acclimation (Table 3). WT and *dewax* showed a 61.5% and 50% increase in CH_2_s, while this functional group was 7.7% lower in *cer3-6* in response to cold acclimation.

Changes in the band intensities in the main lipid regions, represented by CH_3_, CH_2_s, and CH_2_a, in WT, *cer3-6*, and *dewax* in response to cold acclimation, are consistent with the changes of cuticular lipids observed by GC-MS analysis. The spectral profiles of all the genotypes under both control and cold acclimation treatment are documented (Figure 4A and Appendix A).

To understand the distribution of ATR-FTIR data (Table 3), we reduced the dataset into a two-dimensional principal component analysis (PCA) plot. The PCA plot revealed that peak intensity data of the control condition and cold-acclimated genotypes are different from each other, and PC1 and PC2 together accounted for 92.3% of the total variance. As such, PC2 was able to clearly separate components of the control condition genotypes (WT C, dewax C, and cer3-6 C) from those of the corresponding cold-acclimated genotypes into distinct clusters at 95% confidence intervals of the group clustering (Figure 4B). Within the cold-acclimated genotypes, cer3-6A showed a much higher PC1 score.

To determine how the GC-MS data relate to ATR-FTIR spectroscopy data, a correlation heatmap with hierarchical clustering was generated using variables between GC-MS data and ATR-FTIR data (Figure 5; Appendix A). The ATR-FTIR data formed two major clusters, with CO and CH_3_ in one cluster and CC and CH_2_ in the other. GC-MS also formed two major clusters with C-24, 26, 28, 29 in one cluster and all the other longer chain alcohols, alkanes, and fatty acids in a much bigger cluster. When we looked at the correlation of two datasets, C24-, C26-FA waxes had a high positive correlation of 0.88 and 0.91, respectively, with CH_3_. Similarly, C28 FA had a very high positive correlation of 0.98 and 0.99, respectively, with CH_2_a and CH_2_s (Figure 5). In fact, all low-chain FAs (except C32 FA and C34FA) tended to have positive correlation with CH_2_a and CH_2_s, with the two highest peak areas generated in ATR-FTIR analysis of rosette-leaf cuticle in *Arabidopsis*. C29 primary alcohol was found to have positive correlation with the CH_3_ peak area, whereas C30, C32, and C34 PAs were found to be positively correlated with CH_2_a and CH_2_s. Interestingly, alkanes (C29, C31, C33, and C35) did not show any positive correlation with the CH_3_ stretching of ATR-FTIR data. In fact, all GC-MS components showed either no correlation or negative correlation with RCH32s and RCO2s. C29 alkane showed slight positive correlation with CH_2_a and CH_2_s (0.38 and 0.53, respectively, please see Appendix A), and C35 and C31 alkanes showed weak positive correlation to CH_2_s (0.33 and 0.15, respectively). Most alkanes displayed negative correlation with CO. C31 and C33 showed weak positive correlation with CC (Figure 5).

## 3. Discussion

### 3.1. Arabidopsis Requires an Intact Alkane Biosynthetic Pathway, Mediated by CER3 and CER1, to Resist against Dehydration and Frost

Numerous studies have described the role of cuticular wax in connection to drought resistance in plants. In comparison, there are not many studies that have defined the role of cuticular wax in low-temperature and freezing stress and even fewer which have examined both. The present study aimed to examine the role of cuticular wax not only in reducing dehydration water loss but also in freezing avoidance in model plant *Arabidopsis*. A collection of mutants and overexpressing alleles with a defect in cuticular wax production as comparison to their parental WT were utilized in this study. Since changes in the cuticular wax is often connected to cuticular water loss, the dehydration experiment was initiated to assess reduction of water loss of each *Arabidopsis* line. The *cer3-6* mutant allele, reported to be deficient of approximately 83% to total cuticular wax load [15,16], displayed the highest sensitivity to shoot dehydration. The *cer1-4* mutant was not as sensitive as *cer3-6* to dehydration water loss. Both *CER1*- and *CER3*- coded enzymes are involved in the alkane biosynthetic pathway, where CER3 is believed to be involved in converting VLCFA to aldehydes, and CER1 functions to convert aldehydes to alkanes [10,11,12,13]. The knockout mutation in *CER1* generates a deficiency of 44–56% of total wax, which is less than the wax deficiency in *cer3-6* and had less sensitivity to water loss under dehydration. Thus, quantitative reductions appear to be as important as qualitative alterations.

Mutant allele *cer6* also showed sensitivity to water loss under dehydration and contains a stem wax-load of 6–7% of the WT *Arabidopsis* with a significant deficiency of VLC wax component longer than C24 [19]. Interestingly, the double mutant *cer2-5 cer2-like1-1* showed resistance to water loss under dehydration. The *cer2-5 cer2-like1-1* mutant was found to be deficient in wax components longer than C29, although the total wax load in the rosette leaves remained intact in the seedlings with a wild-type level of C29 alkane [20]. C29- and C31- alkanes are two of the most prominent wax components in *Arabidopsis* rosette leaves, and *cer2-5 cer2-like1-1* is deficient of C31 alkane compared to WT. The double mutant has an elevated level of C28 primary alcohols and C28 aldehyde [20]. It is possible that the WT level of overall wax load, along with the dominant presence of C29 alkane and C28 primary alcohols, allowed *cer2-5 cer2-like1-1* to resist the dehydration water loss.

The wax-overproducing mutant *dewax* displayed higher resistance to both dehydration and freezing, confirming the positive effect of high cuticular wax accumulation. A study by the authors of [27] showed that the total wax load in *dewax* was elevated by around 15% with an increase in the C29–33 alkanes, whereas DEWAX OX1 showed a deficiency of total wax-load with decreased C29–33 alkanes and C26–C28 primary alcohols. All these results reconfirm that the presence of hydrophobic wax constituents (i.e., C29–C33 alkanes) in the cuticular wax are important for *Arabidopsis* to resist dehydration water loss.

As a species, *Arabidopsis* is generally treated as frost tolerant. However, there is evidence in the literature this model plant has the ability to avoid freezing [38,39] through the mechanism of supercooling [38]. The freezing phenotype of *dewax* and *cer3-6* clearly indicates cuticular wax can play a role in *Arabidopsis* freezing avoidance. A cuticle with high content of hydrophobic waxes, such as n-alkanes, most likely increases the supercooling ability of *Arabidopsis*. The results also support the prospect that hydrophobic waxes can help the plant to avoid freezing under late spring frost or early fall frost in the Canadian prairies.

### 3.2. Higher Accumulation of Wax in the Cuticle Does Not Inhibit the Reproductive Yield in Arabidopsis

Reports have indicated that altering the cuticular wax can have a negative impact in normal reproductive and vegetative growth in crop plants [40,41,42]. However, there is evidence that drought-tolerance and yield were higher in crops having more cuticular wax than those with less wax or non-waxy crops [42,43,44]. Glaucous wheat genotypes produced significantly more grain than nonglaucous ones in the normal and moderate drought environment [44]. Similarly, the positive correlation between barley grain yield and the epicuticular wax load under drought stress was reported [45]. Data from all these studies also support the idea that it is possible to alter the cuticular wax content and properties (e.g., composition and thickness) without much compensation to the grain yield in crop species. Moreover, loss in crop yield due to various environmental stresses under a continuously changing climate condition can be tackled more efficiently by engineering the desirable epicuticular wax composition.

The lines *cer1-4*, *cer3-6*, *cer6*, DEWAX OX1, and DEWAX OX2 produced abnormal siliques (empty of seeds), a phenotype caused by conditional male sterility in wax-deficient mutant lines grown under low humidity conditions [11,14,19]. By contrast, *dewax*, *cer2-5 cer2-like1-1*, and MYB94 OX produced normal siliques. Among the three, *dewax* even produced higher numbers of siliques than WT, whereas there were no changes in the number of siliques in *cer2-5 cer2-like1-1*, and MYB94 OX compared to WT. Being an overproducer of cuticular wax, MYB94 OX was expected to produce a similar silique number like *dewax*. The expression of *MYB94* is regulated by abiotic stresses and ABA [29]. It is possible that an unknown mechanism of ABA-controlled growth and stress responses is connected to the phenotype expressed by MYB94 OX line. The reproductive phenotypes displayed by *dewax* and MYB94 OX lines need further exploration. However, the silique data displayed by *Arabidopsis* genotypes used in this study suggests an elevated level of cuticular wax will not compromise the reproductive yield and that wax deficiency can be connected to defective seed development.

### 3.3. Cold Acclimation-Induced Compositional Changes of Wax Constituents in the Cuticle

In leaves, nucleation initiates on the leaf surface [36,46,47,48,49,50] and, subsequently, ice propagates into the leaf [51,52,53]. Therefore, particularly in frost-sensitive species, the initial avoidance of frost is critical for survival [54,55]. To explore the connection between the content of the cuticular wax to subzero freezing, we used cold-acclimated *Arabidopsis* wax mutants and overexpressing lines. Cold acclimation increased *dewax* mutant alkane levels compared to WT and reduced alkane levels in *cer3-6.* The *cer3-6* mutant froze at warmer subzero temperatures than WT. Conversely, the cold-acclimated *dewax* mutant froze at cooler subzero temperatures than WT. Thus, the presence of higher cuticular alkane wax-load may enable *Arabidopsis* to avoid freezing for a longer duration under subzero temperatures. According to the authors of [35], drought and high salinity increased fatty acids and alkanes in the cuticular wax loads in *Arabidopsis*. In fact, increased alkane levels are commonly observed drought response phenotypes in most plant species [2]. Moreover, both the acyl decarbonylation and primary alcohol pathways were downregulated under cold acclimation, with a subsequent decrease in total alkanes and total primary alcohols. The *cer3-6* mutant almost entirely lost the ability to produce alkanes under cold acclimation, which infers the presence of a WT copy of *CER3* gene is essential for *Arabidopsis* to keep biosynthesizing alkanes under cold acclimation. Conversely, the *dewax* mutant can keep an elevated level of C29–C35 alkanes even under cold acclimation. The level of C29 alkane in *dewax* did not change compared to WT. The warmer freezing temperature of *cer3-6* and colder freezing temperature of *dewax* clearly indicate that the presence of hydrophobic alkanes is vital for *Arabidopsis* to both avoiding freezing and dehydration stress.

### 3.4. ATR-FTIR Based Analysis Further Revealed Cold-Acclimation Driven Leaf Cuticular Changes in Arabidopsis

The cuticles of cold-acclimated *Arabidopsis* lines were also evaluated using ATR-FTIR. Although GC-MS is the traditional technique to study cuticular wax, the application of ATR-FTIR analysis of cuticular wax has recently been reported [56,57]. The method not only has the capability to identify genotype- and treatment-specific changes in the wax, but also can perform real-time nondestructive analysis of the leaf cuticle. However, compared to GC-MS profile of cuticular wax, the spectroscopic profile generated by FTIR lacks the ability to quantitatively analyze the individual wax constituents of the cuticle. Whereas the *cer3-6* mutant is expected to have a severe reduction (~83%) in the total amount of cuticular wax [16], *dewax* is expected to have a modest increase (~15%) in the wax accumulation [27]. Our results from the GC-MS analysis of selected wax constituents in *cer3-6* and *dewax* agree with the previous studies with these two mutants [15,16,27]. Also, as per the major peak areas that represent lipids in ATR-FTIR spectra, CH_2_a and CH_2_s were higher in the *dewax* and lower in the *cer3-6* under control growth conditions, as expected for cuticular wax analysis. Further, the spectra of CH_2_a and CH_2_s under cold acclimation conditions showed a sharp increase in WT and *dewax* and was unchanged in *cer3-6*. Consistent with this, when the ATR-FTIR spectral profile data was plotted in a two-dimensional PCA plot, PC2 was able to separate cold-acclimated WT and *dewax* and *cer3-6* from the corresponding genotypes under controlled conditions, suggesting that the majority of the variance in the data resulted from the treatments. Thus, GC-MS analysis and ATR-FTIR spectral profile coupled with principle component analysis provides an important platform to elucidate effect of cold acclimation in leaf cuticular wax constituents. Across the three classes of cuticular wax constituents (i.e., fatty acids, alkanes, and alcohols) analyzed by GC-MS, the total wax loads were decreased by 38% and 21% in WT and *dewax*, respectively, under acclimation. The wax loads as per the fatty acids, alkanes, and alcohols was increased by 26% in *cer3-6* under cold acclimation compared to the *cer3-6* grown under control conditions. CH_3_ stretching was increased in all three genotypes (i.e., WT, *cer3-6*, and *dewax*). Therefore, the data indicate that changes in major wax lipid constituents (e.g., VLC fatty acids (C24–34), alkanes (C29–35), and alcohols (C26–34)) under acclimation was not manifested by CH_3,_ CH_2_a, and CH_2_s spectra in FTIR. The correlation study of GC-MS and FTIR data (depicted in Figure 5) also supports the same notion. For example, C29 alkane, one of the most abundant waxes in *Arabidopsis* rosette leaves, had a weak positive correlation with CH_2_a and CH_2_s. However, none of the other alkanes, including the abundant C31 alkane, followed the same trend. It is possible that the contribution to the spectral changes in CH_3_, CH_2_a and CH_2_s under cold acclimation comes from cutin esters and other cuticular constituents. Additional GC-MS analysis may be helpful in confirming our findings. The FTIR spectral data revealed that CO peak areas (1758–1726 cm^−1^), designated as the saturated esters [58], were increased in all *Arabidopsis* genotypes, with a huge peak observed in the leaf cuticle of *cer3-6*. FTIR profiles of lipid peak areas detected by CH_2_a and CH_2_s showed that both WT and *dewax* had a sharp increase in CH_2_a and CH_2_s compared to *cer3-6*. In fact, the sharp increase in the CH_2_a and CH_2_s under cold acclimation was observed for all the genotypes except the *cer3-6* mutant line (Figure 4A and Appendix A). The data reiterate the importance of the CER3 protein in *Arabidopsis* frost avoidance. We also tested the lipid profile of cauline leaves from the plants grown and developed under cold-acclimation conditions (5 °C/2 °C, 16/8 h light-dark cycle at 100 µE m^−2^ s^−1^) in WT, *cer3-6*, and *cer1-4* (Appendix A). Cauline leaves of cold-acclimated WT *Arabidopsis* had a much sharper increase in CH_3_, CH_2_a, and CH_2_s profiles, but this increase was absent in the cauline leaf of the *cer3-6* mutant line. These findings further support that CER3 protein plays an important role in regulating cuticular changes in wax production under cold acclimation.

## 4. Materials and Methods

### 4.1. Plant Materials and Growth Conditions

All mutants and overexpressing lines used in the study are descendants of *Arabidopsis thaliana* (Ecotype: Columbia; Col-0). The mutant lines *cer1-4*, *cer2-5 cer2-like1-1*, *cer3-6*, and *cer6* were confirmed homozygous lines kindly donated by Professor Ljerka Kunst of Botany Department at the University of British Columbia in Canada. The mutant lines *dewax* and transgenic overexpressing lines DEWAX-OX1 and MYB94 OX were generously donated by Professor Mi Chung Suh of Department of Bioenergy Science and Technology, Chonnam National University, Korea. Functional and genotypic characterization of all the genotypes were performed in various previous studies [10,15,16,19,20,27,29]. Plants were grown in 10 × 10 inch pots filled with soilless mix (Sunshine mix #4, Sungro Horticulture, Agawam, MA, USA) supplemented with 1.0 g of 14-13-13 fertilizer. Seeds were surface-sterilized by sequentially soaking in 75% ethanol, rinsing 4–5 times in sterile water, soaking in 20% (*v*/*v*) commercially available bleach for 15 min with stirring, and then rinsing 4–5 times with sterile water. Prior to planting in soil, seeds were placed in sterile 1.5 mL Eppendorf tubes with 250 µL of H_2_O and kept for 3 days in the dark at 4 °C to encourage synchronized germination. Seeds were allowed to germinate and grow inside growth chambers set at 22 °C/20 °C under a 16/8 h light-dark cycle at 100 µE m^−2^ s^−1^.

### 4.2. Arabidopsis Shoot Dehydration Assay

Three-week-old mutants (12–14 rosette leaves) and overexpressing and WT *Arabidopsis* plant lines were used in the whole shoot dehydration assay. All the plants used in the assay were well watered 6 h prior to assay. Shoot sections (including rosettes and stems) from individual plant lines were excised from the root, and the cut-areas were sealed with vacuum grease. Each excised shoot was placed on an analytical balance and allowed to dehydrate under room temperature and in the light. The gravimetric weight loss of each shoot was monitored separately, and videos were recorded for 30 min for each shoot of each genotype. The experiment was repeated 3 times using each of the genotype.

### 4.3. Cold Acclimation

*Arabidopsis* mutants and WT seedlings grown for 2 weeks were cold acclimated at 5 °C/2 °C under a 16/8 h light-dark cycle at 100 µE m^−2^ s^−1^ for 2 weeks prior to freezing treatment, and leaf wax changes were analyzed by ATR-FTIR and GC-MS.

### 4.4. Freezing Treatment and Thermal Imaging

To investigate the role of cuticular wax in plant responses to freezing stress, mutants and overexpressing lines were exposed to subzero temperature treatment following a 2-week regime of cold acclimation of 5 °C/2 °C under a 16/8 h light-dark cycle. The phenotypes of the acclimated plant lines were tested for both freezing tolerance and freezing avoidance. Freezing tolerance analysis consisted of cold-acclimated seedlings nucleated in a programmable freezing chamber (Cincinnati Sub-Zero (CSZ) Temperature Chamber, Cincinnati, OH, USA) at −2 °C and held at each sampling temperature for 4 h, during which the temperature was decreased at a rate of −4 °C/h to −6 °C, 10 °C, −14 °C, and −18 °C and then returned to control growth condition (22 °C/20 °C;16 h/8 h; light/dark) following an overnight incubation at 4 °C. After 7 days of recovery, the survival percentage of each genotype was calculated for all incubation periods. Interestingly, there were no significant differences in the survival percentages observed among all the genotypes (data not shown). Frost avoidance studies were conducted under a quick ramp from 4 °C to −25 °C at a rate of 1 °C/min. The freezing exotherm of *Arabidopsis* plants was measured using an infrared thermal video/image capturing camera FLIR T640BX (Manufactured in 2013, FLIR System Inc., Wilsonville, Oregon, USA) according to the methodology described by the authors of [55]. The camera was programmed in continuous auto adjustment as per the temperature span of the freezing treatment. The camera had 640 × 480-pixel resolution and 0.035 °C sensitivity. Images and video from the camera were processed and analyzed using FLIR ThermaCam Researcher Pro 2.8. Plants were not ice nucleated, and the root zone did not freeze in the frost avoidance studies.

### 4.5. ATR-FTIR Analysis of Rosette Leaf Cuticular Waxes

In the ATR-FTIR spectroscopy analysis of *Arabidopsis* cuticular wax, fresh rosette, or cauline leaves collected from 3–5 individual cold-acclimated and non-acclimated (control) plants of each genotype were used. The ATR-FTIR spectroscopy analysis was performed at the Canadian Light Source (CLS) facility in Saskatoon, Saskatchewan. The ATR end station at the CLS facility was a Pike MiracIATR with a 45° Ge ATR Crystal equipped with a deuterated-triglycine sulfate (DTGS) detector at room temperature. Pike MiracIATR probe accessory (radius of 3 mm) was placed on the *Arabidopsis* leaf adaxial or abaxial surfaces, where the infrared light was generated through a Globar equipped with a silicon carbide source. Spectra were obtained in the range of 800–4000 cm^−1^ with an average of 512 scans per spot that was normalized to background scan of an empty probe. Finally, the resulting spectra of each *Arabidopsis* genotype were processed and analyzed in Orange (v3.3.10 Open source). The infrared peak areas for lipids were taken into consideration for the changes of leaf cuticular wax in response to acclimation. Integrations for the peak areas were performed on 5 regions: CH_3_ stretching (asymmetrical + symmetrical) was assigned to 2966–2950 cm^−1^, asymmetrical CH_2_ was assigned to 2936–2894 cm^−1^, symmetrical CH_2_ was assigned to 2871–2826 cm^−1^, CO stretching was assigned to 1758–1726 cm^−1^, and CC stretching was assigned to 1706–1584 cm^−1^.

### 4.6. Correlation and Principal Component Analysis

To study the relationship between GC-MS and ATR-FTIR data, the Pearson correlation was calculated from the datasets that included 16 variables of GC-MS and 8 variables of ATR-FTIR on 6 genotypes. Correlation heatmap was generated using the pheatmap function in R package (https://cran.r-project.org/web/packages/pheatmap, accessed on 9 October 2020). Principal component analysis (PCA) for ATR-FTIR data was carried out using the online clustVis tool [59].

### 4.7. Wax Extraction and GC-MS Analysis

Wax extraction and analysis were performed as per the protocol developed and illustrated by the authors of [60] with minor modifications [20,60]. In general, the chemical analysis of the extracted wax involved the quantitation of wax monomers by gas chromatography coupled with flame ionization detection (GC/FID) and the identification of wax monomers by mass spectrometry. Rosette leaf samples of non-acclimated and cold-acclimated *Arabidopsis* seedlings were freshly collected from WT, *cer36*, and *dewax*, and dipped in 10 mL chloroform with internal standard (10 µg tetracosane (C24 hydrocarbon) in 100 µL chloroform). Leaf area measurement was performed as per the methodology described by the authors of [60]. The extracted samples were dried under a nitrogen stream. Samples were transferred to an autosampler vial in 150 µL chloroform and dried under nitrogen. Then, 50 µL pyridine, N,O-Bis(trimethylsilyl)trifluoroacetamide (BSTFA), and Trimethylchlorosilane (TMCS) (50:49:1) were added, and the vial was sealed and heated to 80 °C for 1 h to derivatize the alcohols and fatty acids to their TMS derivatives to facilitate chromatography. Samples were run on an Agilent 7890N gas chromatograph equipped with an HP-1 column (Agilent 19091Z-313) and run under the conditions described by the authors of [60] and connected to either a flame ionization detector or an Agilent 5973 Mass Selective Detector that was run under standard electron impact conditions (70 eV), scanning an effective *m/z* range of 40–700 at 2.26 scans/s and tuned to the “standard tune” profile used for the NIST08 mass spectral library (Agilent G1035B).

Mass spectral library searches using the NIST08 library as well as the expected elution order were used to determine the compounds of interest in our samples. Reference materials (e.g., purified wax constituents obtained from Sigma-Aldrich Canada, Oakville, ON, Canada) were used as calibration standards to determine correction factors for the quantitation of TMS derivatives of the alcohols and fatty acids. The correction factor of the hydrocarbons to the internal standard was assumed to be 1.00 on a weight basis. The calibration standard solutions and the samples were then run on the GC/FID for accurate quantitation of each compound of interest. The peak profile, elution order, and relative retention times from MS data of representative samples for each of the *Arabidopis* lines were used to identify FID peaks for all the samples analyzed.

## 5. Conclusions

Using several wax mutants and overproducing lines, the present study explored the role of cuticular wax in *Arabidopsis* adaptation to shoot dehydration and low-temperature stress. *Arabidopsis* wax deficient genotypes *cer1-4*, *cer3-6*, cer6, and DEWAX OX1 showed more sensitivity to shoot dehydration, whereas wax-overproducing mutant *dewax* was more resistant. At subzero freezing conditions, *cer3-6* showed freezing exotherms at warmer temperatures, and *dewax* froze at colder temperatures compared to WT. Cuticular wax positively impacted the ability of *Arabidopsis* to resist shoot dehydration and freezing. Further analysis of the changes in cuticular waxes in response to cold acclimation revealed that the absence of the *CER3* gene in *cer3-6* mutant line can completely diminish its ability to biosynthesize VLC alkanes, the dominant hydrophobic wax constituents of the cuticle. *CER3,* along with the acyl decarbonylation pathway gene *CER1*, have been treated as evolutionarily conserved genes, playing important roles in the biosynthesis of wax and modulating the drought stress response [61]. Various orthologs of CER3 proteins are found in other important crop plant species. For example, there are three close orthologs for *Arabidopsis* CER3 in *Triticum* A genome: TraesCS5A02G220200, TraesCS6A02G150000, and TraesCS7A02G501400 on chromosome 5, 6, and 7, respectively, with 72–75% identity with AtCER3 at the protein level. ZmGL1 in *Zea* and OsWSL2 in *Oryza* share 62% and 64.46% protein identity with AtCER3, respectively. Although *cer1-4* was not as sensitive as *cer3-6* in our studies, recently, the ortholog of *CER1* in wheat, *TaCER1-1A,* was reported to have significantly reduced levels of C33 alkanes in wheat, while overexpression in rice resulted in elevated levels of C25–C33 alkanes compared to the wild type [62]. We provide further evidence that the role of the CER3 protein can be extended beyond dehydration stress, with a role in frost avoidance. The orthologs of the *CER3* can be a potential target for gene editing to generate climate resilient crops.

## Figures and Tables

**Figure 1 ijms-22-01554-f001:**
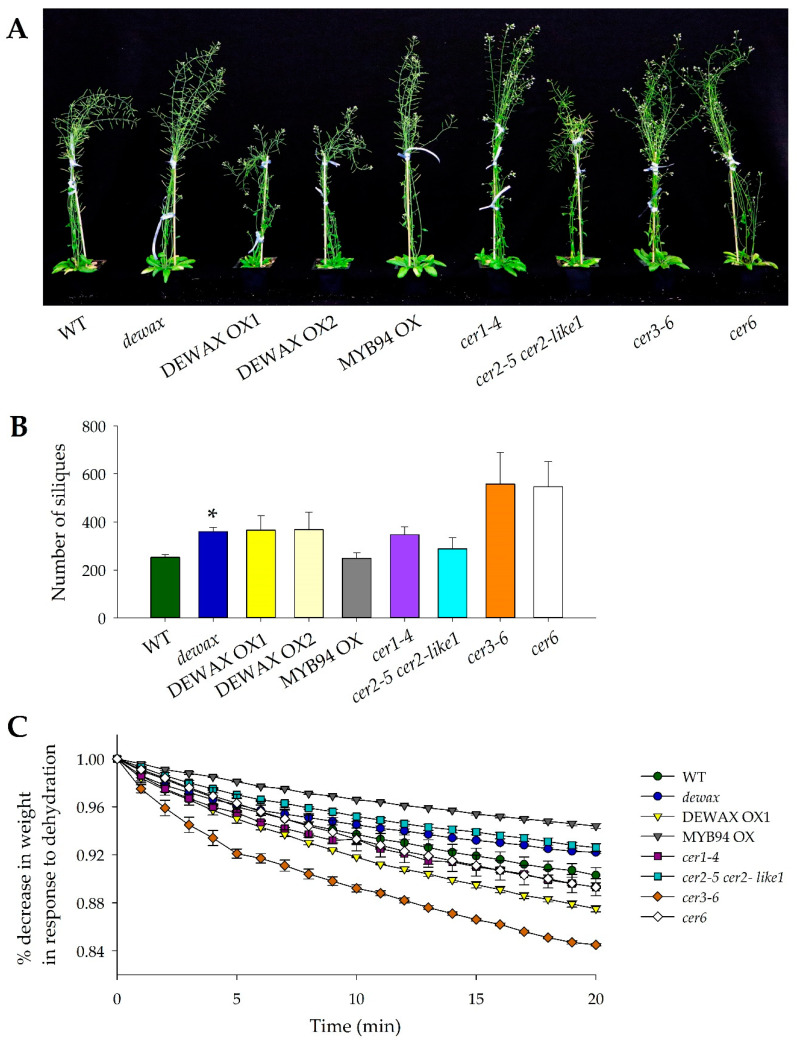
Phenotype of the *Arabidopsis* lines. (**A**) Seven-week-old *Arabidopsis* plants grown inside a growth chamber set at 22 °C/20 °C under a 16/8 h light-dark cycle at 100 µE m^−2^ s^−1^ (no humidity control). (**B**) Number of siliques generated per plant at maturity for each *Arabidopsis* line. Bar diagrams represent the average of three replicates with standard error (SE). Student’s *t*-test was run to look for the differences in the total number of siliques at maturity between wild-type (WT) and *dewax*, MYB94 OX, and *cer2-5 cer2-like1* lines. The thresholds of significance are indicated above the histogram (* *p* < 0.05). (**C**) Water loss (as represented by the time-course decrease in weight of 3-week-old plants) in response to dehydration stress. Shoot section of the plants were used in the assay. Average weight loss/min for three plants with SE is presented.

**Figure 2 ijms-22-01554-f002:**
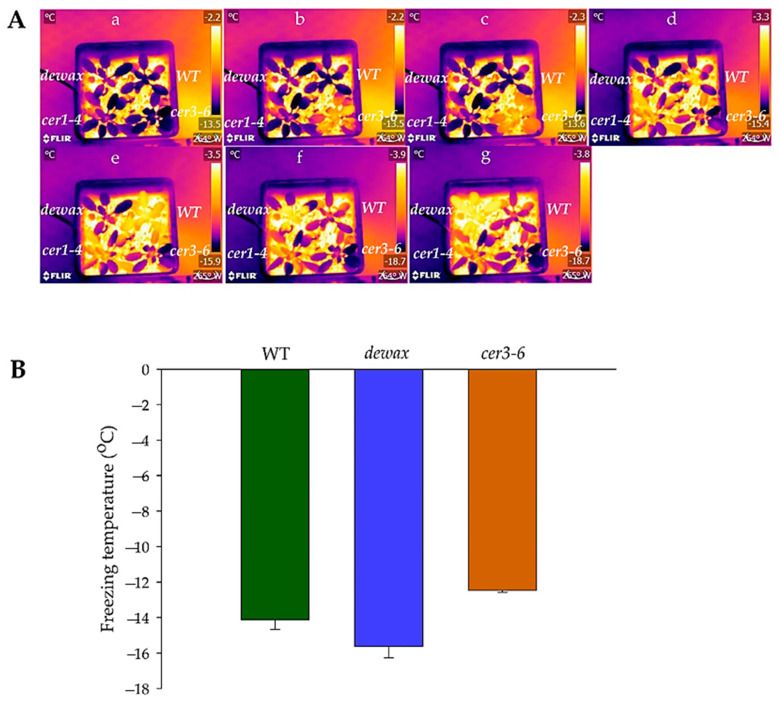
Responses of cold-acclimated *Arabidopsis* lines WT, *cer3-6*, and *dewax* together in the rapid freezing treatment. (**A**) Infrared video-captured thermal images of cold-acclimated WT, *dewax*, and *cer3-6* lines indicate which plants froze under rapidly decreasing subzero conditions (−2C to −15C in 13 min) in which soil remained unfrozen across seven sequential time points. False color thermal bar (right side of each panel) indicates those plants which were initially supercooling (dark purple) and those which had subsequently just frozen, releasing an exotherm (yellow to white) with subsequent cooling of the frozen plant (dark orange to purple). **a**: Before ice nucleation started in *cer3-6*; **b**: Start of ice nucleation in *cer3-6*; **c**: Five seconds after the start of ice nucleation in *cer3-6*; **d**: Start of the ice nucleation in WT; **e**: Five seconds after the initiation of ice nucleation in WT; **f**: Start of ice nucleation in *dewax*; **g**: Five seconds after the start of ice nucleation in *dewax*. (**B**) Lethal freezing temperature at which cold-acclimated WT, *dewax* and *cer3-6* first displayed ice nucleation. Tests were repeated two times to compare *dewax* and *cer3-6* with the WT in the same experiments. Bar plots represent the average of lethal freezing temperatures with SE for all replicates.

**Figure 3 ijms-22-01554-f003:**
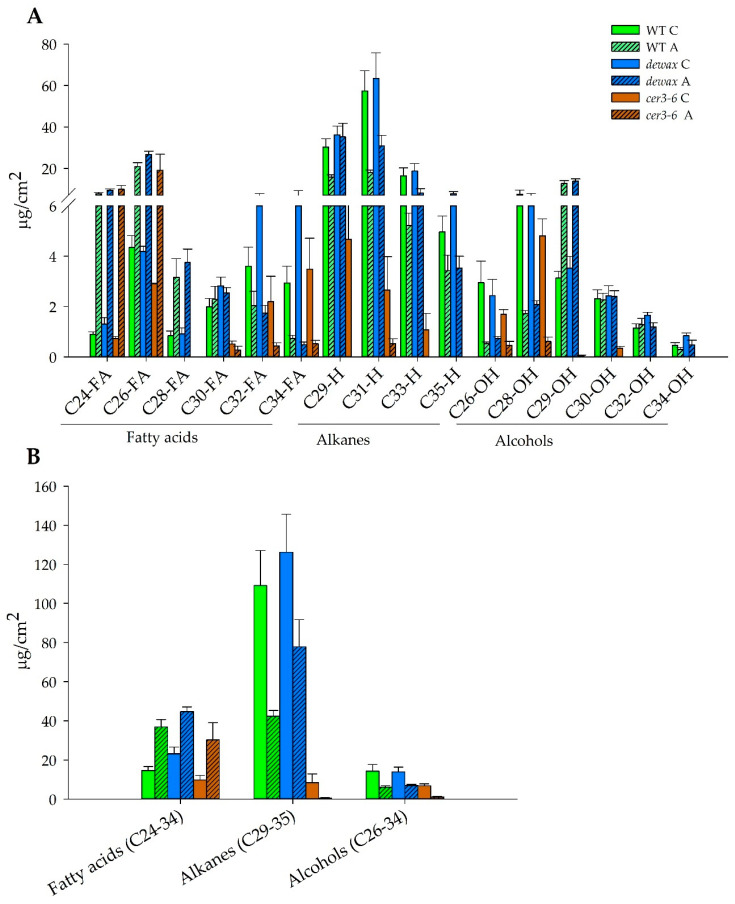
Gas chromatography Mass spectrometry (GC-MS) analysis of wax constituents in WT, *dewax*, and *cer3-6* in non-acclimated (C) and cold-acclimated (A) *Arabidopsis* lines. (**A**) Rosette leaves extracted wax constituents of various chain-length under control and cold-acclimated conditions. (**B**) The total amount of fatty acids, alkanes, and alcohols under control and cold-acclimated conditions in rosette leaves. Bar diagrams represent the average of various constituents quantified from four to nine biological replicates of wax samples extracted from rosette leaves of WT, *dewax* and *cer3-6*. Error bars represent the standard error (SE) of mean for 4–9 replicates.

**Figure 4 ijms-22-01554-f004:**
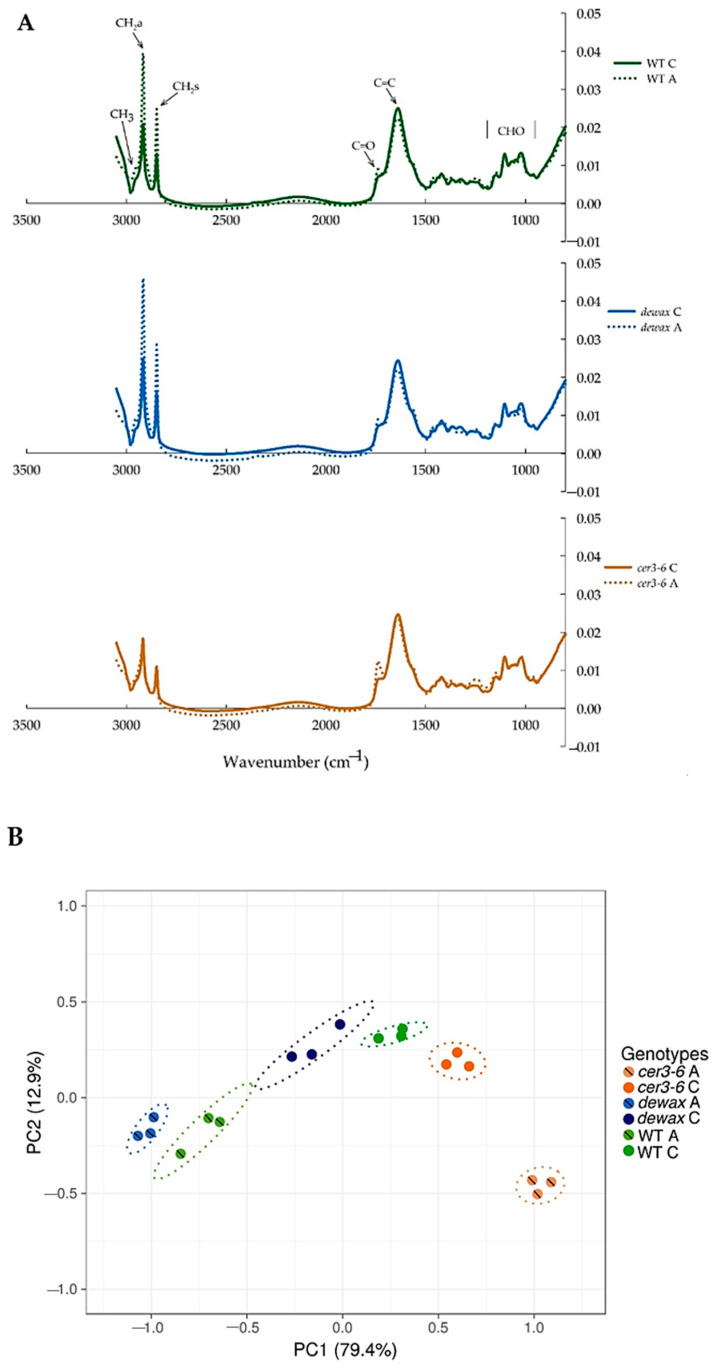
ATR-FTIR analysis of rosette leaf cuticle. (**A**) ATR-FTIR spectral profiles of rosette leaf adaxial surfaces under control (C; solid lines) and cold-acclimated (A; dotted lines) conditions for WT, *dewax*, and *cer3-6* are presented. ATR-FTIR spectral profiles for all remaining genotypes are in the Appendix A. (**B**) Principal Component Analysis (PCA) plot for ATR-FTIR data. Original values were ln(x + 1)-transformed. Pareto scaling was applied to rows, and SVD with imputation was used to calculate principal components. X and Y axis show principal component 1 and principal component 2, explaining 79.4% and 12.9% of the total variance, respectively. Confidence level for ellipses: 0.95. Data points indicate sample replicates.

**Figure 5 ijms-22-01554-f005:**
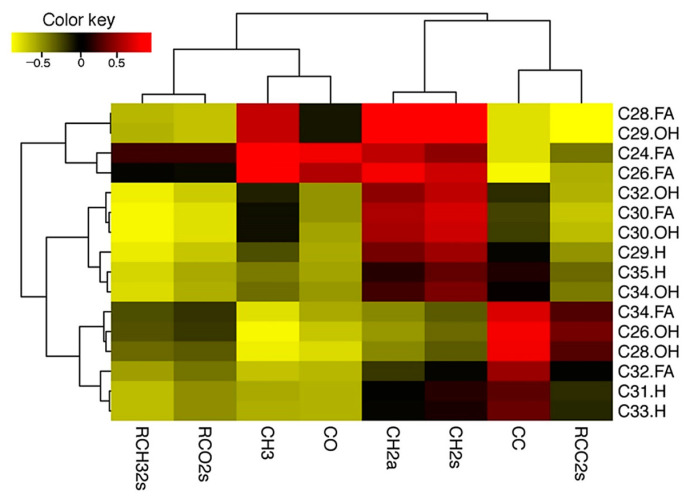
Correlation heatmap with hierarchical clustering between GC-MS data and ATR-FTIR data. Yellow: Negative correlation; Red: Positive correlation; Clustering distance: Euclidean; Method: Ward methodology; Rows: FTIR variables; Columns: GC variables.

**Table 1 ijms-22-01554-t001:** Nucleation temperatures of cold-acclimated *dewax* and *cer3-6* relative to the cold-acclimated WT ^1^.

Genotype	Relative Nucleation Temperature
WT	1
*dewax*	1.14 ± 0.06 *
*cer3-6*	0.87 ± 0.04 *

^1^ Tests were repeated three to four times to compare *dewax* and *cer3-6* with the WT in separate experiments. Values represent the average relative nucleation temperature with SE for all replicates. Student’s *t*-test indicate the significance differences (* *p* < 0.05).

**Table 2 ijms-22-01554-t002:** Two-way ANOVA of the changes in wax constituents in WT, *dewax*, and *cer3-6* in response to cold acclimation.

**Fatty Acids (FA)**								
		**C24-FA**	**C26-FA**	**C28-FA**	**C30-FA**	**C32-FA**	**C34-FA**	**Total FA**
		**Least square means (LSM)**						
Treatment (T)								
Non-acclimated (C)		0.98	3.81	0.59	1.78	4.26	4.44	15.86
Cold-acclimated (A)		9.01	22.26	2.30	1.70	1.41	0.59	37.26
Genotype (G)								
WT		4.33	12.65	2.00	2.13	2.82	1.84	28.76
*dewax*		5.34	15.49	2.33	2.68	4.37	3.70	33.91
*cer3-6*		5.32	10.97	0.00	0.39	1.32	2.01	20.01
Analysis of variance								
Source	DF				Pr ≥ *F*			
T	1	<0.001	<0.001	<0.001	n.s. ^1^	<0.001	<0.001	<0.001
G	2	n.s.	n.s.	<0.001	<0.001	0.01	n.s.	0.02
T × G	2	n.s.	n.s.	0.03	n.s.	n.s.	n.s.	n.s.
**Alkanes (H)**								
		**C29-H**	**C31-H**	**C33-H**	**C35-H**	**Total H**		
		**LSM**						
Treatment (T)								
C		23.72	41.19	12.09	4.23	81.23		
A		16.98	16.47	4.47	2.32	40.23		
Genotype (G)								
WT		23.00	37.66	10.84	4.20	75.71		
*dewax*		35.71	47.24	13.44	5.62	102.01		
*cer3-6*		2.33	1.60	0.54	0.00	4.47		
Analysis of variance								
Source	DF				Pr ≥ *F*			
T	1	0.04	<0.001	0.004	0.002	0.002		
G	2	<0.001	<0.001	<0.001	<0.001	<0.001		
T × G	2	n.s.	0.05	n.s.	0.03	n.s.		
**Alcohols (OH)**								
		**C26-OH**	**C28-OH**	**C29-OH**	**C30-OH**	**C32-OH**	**C34-OH**	**Total OH**
					**LSM**			
Treatment (T)								
C		2.36	6.28	2.23	1.70	0.93	0.43	13.94
A		0.58	1.47	8.83	1.56	0.83	0.26	13.53
Genotype (G)								
WT		1.74	4.59	7.92	2.30	1.22	0.38	18.13
*dewax*		1.59	4.33	8.67	2.42	1.42	0.67	19.08
*cer3-6*		1.08	2.71	0.02	0.18	0.00	0.00	3.98
Analysis of variance								
Source	DF				Pr ≥ *F*			
T	1	0.002	<0.001	<0.001	n.s.	n.s.	n.s.	n.s.
G	2	n.s.	n.s.	<0.001	<0.001	<0.001	<0.001	<0.001
T × G	2	n.s.	n.s.	<0.001	n.s.	n.s.	n.s.	n.s.

^1^ n.s., not statistically significant (*p* > 0.05).

**Table 3 ijms-22-01554-t003:** Attenuated Total Reflection Fourier Transform Infrared Spectroscopy (ATR-FTIR) spectroscopic characteristics of rosette leaf adaxial cuticle surfaces of WT, *dewax*, and *cer3-6* (peak area integration and area ratios).

	CH_3_ Stretching (2966–2950)	CH_2_ Asymmetric (2936–2894)	CH_2_ Symmetric (2871–2826)	C=O Stretching (1758–1726)	C=C Stretching (1706–1584)	Ratio (CH_3_:CH_2_s)	Ratio (C=O:CH_2_s)	Ratio (C=C:CH_2_s)
	CH_3_	CH_2_a	CH_2_s	CO	CC	RCH32s	RCO2s	RCC2s
WT C	0.052 ± 0.027	0.27 ± 0.05	0.138 ± 0.025	0.109 ± 0.002	1.143 ± 0.015	0.378	0.791	8.310
WT A	0.077 ± 0.009	0.462 ± 0.039	0.216 ± 0.047	0.132 ± 0.007	1.023 ± 0.02	0.358	0.612	4.729
*dewax C*	0.051 ± 0.032	0.303 ± 0.07	0.162 ± 0.037	0.111 ± 0.005	1.127 ± 0.032	0.316	0.688	6.966
*dewax A*	0.07 ± 0.008	0.517 ± 0.019	0.24 ± 0.009	0.128 ± 0.003	1.014 ± 0.027	0.311	0.536	4.234
*cer3-6 C*	0.06 ± 0.014	0.255 ± 0.014	0.128 ± 0.01	0.116 ± 0.008	1.134 ± 0.02	0.472	0.912	8.889
*cer3-6 A*	0.069 ± 0.011	0.277 ± 0.014 * (*p* = 0.024); WT A vs. *cer3-6 A*)	0.118 ± 0.019 * (*p* = 0.021); WT A vs. cer3-6 A)	0.174 ± 005 ** (*p* = 0.008); WT A vs. *cer3-6 A*)	1.081 ± 0.023	0.588	1.477	9.183

* *p* < 0.05; ** *p* < 0.01.

## Data Availability

Data are available upon request.

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
