# Peer review of "Dissecting the Roles of Cuticular Wax in Plant Resistance to Shoot Dehydration and Low-Temperature Stress in Arabidopsis"

_ijms, 2021, doi:10.3390/ijms22041554_

Round 1

Reviewer 1 Report

The manuscript has improved and I now find it acceptable for publication if the current Fig. 2 is replaced with Fig. S1 with more replicates (see below). 

I am very glad to see Fig. S1, where the freezing exotherms of the WT and the mutant lines were determined in the same runs. I find Fig. S1 much more interesting and convincing than Fig. 2 and I suggest to include Fig. S1 in the manuscript, while moving the current Fig. 2 to supplementary material (or omit it). This may require some further replicates though (the current Fig. S1 is based on two replicates vs. 3-4 reps in the current Fig. 2).

The authors have included new statistical tests of the GC-MS data (Fig. 3). As far as I understand these tests are two one-way ANOVAs to test whether wax constitutes differ between non-acclimated genotypes and cold acclimated genotypes. However, I think it would be even more useful to analyze the data using a two-way ANOVA with genotype and treatment (non-acclimated and cold acclimated) as factors. Then it is possible to test for interactions.

Fig. 4B works much better with more data points included as suggested by the other reviewer.

Author Response

(Reviewer comments are in italic bold and our responses are in regular font)

The manuscript has improved and I now find it acceptable for publication if the current Fig. 2 is replaced with Fig. S1 with more replicates (see below).

I am very glad to see Fig. S1, where the freezing exotherms of the WT and the mutant lines were determined in the same runs. I find Fig. S1 much more interesting and convincing than Fig. 2 and I suggest to include Fig. S1 in the manuscript, while moving the current Fig. 2 to supplementary material (or omit it). This may require some further replicates though (the current Fig. S1 is based on two replicates vs. 3-4 reps in the current Fig. 2).

 We replaced  Figure 2 with Figure S1 in the updated manuscript. We have also added a Table (please refer to Table 1 in the updated manuscript) which reflects  3 – 4 replications of  IR exotherm data  of cold-acclimated dewax and cer3-6 relative to the cold-acclimated WT. The relative presentation of freezing data (present in Table 1)  is consistent with the results of the new Fig. 2.

(All the updated changes can be found in lines 190-223,  Figure 2, and in Table 1.)

The authors have included new statistical tests of the GC-MS data (Fig. 3). As far as I understand these tests are two one-way ANOVAs to test whether wax constitutes differ between non-acclimated genotypes and cold acclimated genotypes. However, I think it would be even more useful to analyze the data using a two-way ANOVA with genotype and treatment (non-acclimated and cold acclimated) as factors. Then it is possible to test for interactions.

 Two-way ANOVA test data is included now. Please refer to Table 2 of the updated manuscript.

 Fig. 4B works much better with more data points included as suggested by the other reviewer".

We are keeping Fig. 4B as is. Thank you!

Reviewer 2 Report

All of the comments form both of the reviewers have been considered/answered and implemented where appropiate based on the authors competencies.

Author Response

There were no additional revisions requested from Reviewer 2.

This manuscript is a resubmission of an earlier submission. The following is a list of the peer review reports and author responses from that submission.

Round 1

Reviewer 1 Report

Dear Authors,

Thank you for the opportunity to review this fine article! Experiments have been designed in a way that statistical conlcusions could be drawn, the relations between the wax deficient, overexpressed mutants and wax composition is ellegantly demonstrated. Clear differences can be seen between the different mutant lines and WT indicating the importance of cuticular wax for the resistances investigated in this study. The obtained results are conclusive and the possible application potential of the findings are clearly indicated by the authors. The article is well written in good English and require very minimal spelling check. This is a very good article that will be basis of studies dealing with similar questions for the same reasearchers or possibly other scientific groups!

Introduction

Intoduction explains the known and the not know about the topic. The aim of the study is clearly stated. Detailed explanation of the functions of each gene helps the reader to immerse into the topic. Both the genetics and chemical composition of Arabidopsis wax are well explained and the connection between the two is clearly written. Very good introduction!

Line 50: aldehyde ->aldehydes

Line 134: check the capitalisation of GC-MS

Results

Well written headings of results subsections. Figures are very well made and the conducted experiments show deep understanding of the performed work, also the captions are detailed and easy to follow. The results of the experiments are well structured into graphs and figures so that the main findings of the study could be understood by looking only at the figures.

Line 145: Could the humidity in the chamber have influence on the result?

Figure 1B: Perhaps axis lines could be added to the graph to be consistent with other graphs.

Figure 1C: Missing the DEWAX OX 2 weight loss curve.

Figure 3: add lines to the axis as in previous graphs. Also previous bar graphs are with black bars (perhaps in this figure black and grey could be used or black and outlined white bar). values on Y axis could be made larger. 

In other studies (although different materials) also other groups of wax constituents were identified, particularly sterols and triterpenoids (https://doi.org/10.3390/foods9050587 ; https://doi.org/10.1016/j.foodchem.2019.05.134 ). Were those compounds not found in the Arabidopsis wax?

Figure 4: More points in the PCA loadings plot would make a more conclusive analysis - it is hard to identify clustering with just a few points, however, based on the FTIR and other analysis it is clear that there are signifficant differences.

Discussion

Important points are made in the discussion that could have implications in the use of the presented results in other model organisms. Very good discussion and summary of findings! The structure of each of the subsections are constant and easy to read which helps in navigating the text. Very good!

Materials and Methods

Materials and methods section is well written and using the information provided it would be possible to reproduce the performed experiments, no informations has been witheld. Some minor language corrections should be made, there are a few extra words or incomplete words in the text.

Line 436: the ->then

Line 448: from? where is to?

Line 451-452: for two weeks....for two weeks

Subsection 4.5: were the FTIR measurements done directly on the leaf surface or extracts (washings) of the cuticular wax?

Subsection 4.6: please follow the citation guidelines for citing R and the packages.

Subsection 4.7: It would have been useful to also perform transesterification of wax constituents. In the mentioned references above there are various groups of compounds found in the wax using direct identification (as you have done) as well as TE procedure. Please mention the used mass spectral library. Also retention indices could be used to help with the identification. No information on the used GC or MS setting (could also be refferenced from other articles).

Conclusions

Good summary with the most important findings, also the future prospects and possible application of the obtained results are indicated. Excellent work!

Reviewer 2 Report

Rahman et al. investigated the role of cuticular wax in reducing low-temperature and dehydration stress in plants using Arabidopsis thaliana mutants and transgenic genotypes with changes in the formation of cuticular wax. The importance of wax in protection against non-stomatal water loss is well established, but to my knowledge, it has not previously been investigated whether changes in cuticular wax plays a role in low-temperature tolerance of plants. Hence, the topic is interesting from a fundamental point of view and most of the data are interesting. The paper is  well written, based on clear methodological approaches and contains well-presented results. Still, I have a number of reservations against data  interpretation and discussion of the results and some of the conclusions.

According to Figs. 2C-D the mean freezing temperature of the wild-type differs by about 2°C in the two test runs (ca. -12°C and -14°C in Fig. 2C and Fig. 2D, respectively). This is approximately the same difference in freezing temperature found between dewax and cer3-6, which according to the authors differ in supercooling ability. Clearly, you would expect the data for the wild-type to be reproducible, and since they are not, I am not convinced that the freezing temperature of dewax and cer3-6 is indeed different. The authors need to make sure they are using a methodology that provide consistent results for the same genotype tested under the same conditions. Alternatively, in addition to the data already presented, they need to test all three genotypes (WT, dewax, cer3-6) or at least dewax and cer3-6 at the same time in order to allow a direct comparison.

According to the results section, freezing tolerance of cold acclimated plants did not differ (based on survival percentage), whereas the freezing exotherm differed (although as indicated above I am not fully convinced), indicating differences in freeze avoidance ability. This requires a discussion of whether Arabidopsis survive freezing by freeze avoidance or freeze tolerance and/or under which environmental conditions freeze avoidance vs freeze tolerance is most important. I am not an expert on Arabidopsis, but in the literature, it is often considered freeze tolerant. Thus, I am wondering whether freeze avoidance is ecologically important in Arabidopsis? Or whether it is mostly of importance when freezing is not externally nucleated and occurs very fast (in the lab)? 

In the conclusion it says “At sub-zero freezing conditions, cer3-6 was more sensitive and dewax was more resistant than WT”. However, according to the survival assay this is not right. Thus, the authors need to be more specific.

One of the conclusions of the study is that higher accumulation of wax in the cuticle does not inhibit the reproductive yield in Arabidopsis. However, I find this conclusion far-ranging as it is only based on quantification of silique numbers and not seed yield (seed no and/or seed mass).

Minor issues:

The introduction is long. I think some of the text concerning wax related genes and wax biosynthesis can be shortened, without losing too much information.

As far as I understand, the authors do not statistically test whether cold acclimation cause a change in wax constituents (GC-MS data)? They only test whether the concentration of different constituents differ in the mutants compared to the wild-type. However, there is an entire paragraph in the discussion focusing on cold acclimation-induced compositional changes of wax constituents. This part of the discussion needs to be supported by statistical tests. 

More of the data are analyzed using Student´s t-test to evaluate potential differences between WT and mutant genotypes. When performing multiple t-tests the authors need to correct for multiple testing. Also, it might be interesting no only to compare the wild-type and one mutant genotype at a time, but also to compare the mutant genotypes.